# High-Resolution Computed Tomography (HRCT) Reflects Disease Progression in Patients with Idiopathic Pulmonary Fibrosis (IPF): Relationship with Lung Pathology

**DOI:** 10.3390/jcm8030399

**Published:** 2019-03-22

**Authors:** Elisabetta Cocconcelli, Elisabetta Balestro, Davide Biondini, Giulio Barbiero, Roberta Polverosi, Fiorella Calabrese, Federica Pezzuto, Donato Lacedonia, Federico Rea, Marco Schiavon, Erica Bazzan, Maria Pia Foschino Barbaro, Graziella Turato, Paolo Spagnolo, Manuel G. Cosio, Marina Saetta

**Affiliations:** 1Department of Cardiac, Thoracic, Vascular Sciences and Public Health, University of Padova and Padova City Hospital, 35128 Padova, Italy; ecocconcelli@icloud.com (E.C.); Elisabetta.balestro@aopd.veneto.it (E.B.); dav.biondini@gmail.com (D.B.); fiorella.calabrese@unipd.it (F.C.); federica.pezzuto@libero.it (F.P.); federico.rea@unipd.it (F.R.); marco.schiavon@unipd.it (M.S.); erica.bazzan@unipd.it (E.B.); graziella.turato@unipd.it (G.T.); paolo.spagnolo@unipd.it (P.S.); 2Institute of Radiology, Department of Medicine, University of Padova, 35128 Padova, Italy; giulio.barbiero@aopd.veneto.it; 3Istituto Diagnostico Antoniano - Affidea, 35100 Padova, Italy; rpolve@libero.it; 4Department of Medical and Surgical Sciences, University of Foggia, Policlinico “OO. Riuniti”, 71122 Foggia, Italy; donatolacedonia@gmail.com (D.L.); mariapia.foschino@unifg.it (M.P.F.B.); 5Meakins Christie Laboratories, Respiratory Division, McGill University, 65591 Montreal, QC, Canada; Manuel.cosio@mcgill.ca

**Keywords:** HRCT, disease progression and lung pathology

## Abstract

High-Resolution Computed Tomography (HRCT) plays a central role in diagnosing Idiopathic Pulmonary Fibrosis (IPF) while its role in monitoring disease progression is not clearly defined. Given the variable clinical course of the disease, we evaluated whether HRCT abnormalities predict disease behavior and correlate with functional decline in untreated IPF patients. Forty-nine patients (with HRCT_1_) were functionally categorized as rapid or slow progressors. Twenty-one had a second HRCT_2_. Thirteen patients underwent lung transplantation and pathology was quantified. HRCT Alveolar (AS) and Interstitial Scores (IS) were assessed and correlated with Forced Vital Capacity (FVC) decline between HRCT_1_ and HRCT_2_. At baseline, AS was greater in rapids than in slows, while IS was similar in the two groups. In the 21 subjects with HRCT_2_, IS increased over time in both slows and rapids, while AS increased only in rapids. The IS change from HRCT_1_ to HRCT_2_ normalized per month correlated with FVC decline/month in the whole population, but the change in AS did not. In the 13 patients with pathology, the number of total lymphocytes was higher in rapids than in slows and correlated with AS. Quantitative estimation of HRCTs AS and IS reflects the distinct clinical and pathological behavior of slow and rapid decliners. Furthermore, AS, which reflects the immune/inflammatory infiltrate in lung tissue, could be a useful tool to differentiate rapid from slow progressors at presentation.

## 1. Introduction

Idiopathic pulmonary fibrosis (IPF) is a chronic and progressive lung disease of unknown origin with a highly heterogeneous and unpredictable clinical course [1,2]. While in most cases the inexorable decline in lung function and symptoms occur over a period of years (slow progressors), 10–15% of individuals experience a much faster course, progressing from mild symptoms to respiratory failure and death over a period of months (rapid progressors) [1,3,4].

The identification of Usual Interstitial Pneumonia (UIP) pattern by High Resolution Computed Tomography (HRCT) plays a central role in the diagnosis of IPF, avoiding the need of lung biopsies in a high proportion of cases [2,5]. Furthermore, HRCT-derived scores for fibrosis extent have been widely shown to correlate with degree of physiological impairment and may be more sensitive to subtle changes in disease status than physiological metrics [6,7,8,9,10,11]. 

Although the crucial role of HRCT in staging and monitoring IPF over time has been emphasized [12], the big challenge for clinicians remains the possibility to forecast the disease course (slow or rapid) at the time of diagnosis [13,14,15]. To predict the variable and poorly defined natural history of IPF, composite scoring systems are increasingly being developed [6,7,16]. The Gender, Age, Physiology (GAP) index, which is based only on clinical and functional variables, was able to predict one-year mortality in a cohort of patients with IPF [16]. Moreover, integrating CT scores to the GAP model increased the accuracy of mortality prediction [7], indicating a potential role for the HRCT in the prediction schemes. However, these scoring systems, even if useful, are still neither able to foresee prospectively the highly heterogeneous and unpredictable disease behavior, nor able to guide treatment response. 

We thought that a careful evaluation of the different HRCT patterns, including not only fibrotic changes but also ground glass opacities, might help to predict the future rate of functional decline in patients with IPF not conditioned by antifibrotic treatment. Taking advantage of our unique population of IPF patients not yet treated with antifibrotics, in this study, we assessed if HRCT pattern at diagnosis may: (a) predict disease behavior (slow or rapid progressors); (b) have a pathological basis; and (c) if changes of the HRCT pattern over time are linked to functional decline, without the confounding factor of treatment. 

## 2. Methods

### 2.1. Study Population

In this longitudinal study, we analyzed a well-characterized cohort of Idiopathic Pulmonary Fibrosis (IPF) patients, with a long clinical functional and radiological follow up, referred between 2011 and 2014, naïve of antifibrotics.

All patients in our study, whether from our center or referred to our center, were offered antifibrotic therapy as soon as it became available, provided they met the Forced Vital Capacity (FVC), DL_CO_ and age criteria for treatment and they had no clear contraindications to it. However, given that the aim of our study was to look at a population of patients off treatment, we considered only radiological and functional data before antifibrotic therapy was instituted. In addition, a minority of our patients belonged to an historical cohort from the pre-antifibrotic therapy era (before 2014) and they had no access to antifibrotic therapy. 

Forty-nine patients from two Interstitial Lung Disease Centres in Italy (University of Padova, Italy, n = 43 and University of Foggia, Italy n = 6) were included. 

The diagnosis of IPF was made in accordance with the latest guidelines [1,2] (Appendix A). Clinical and functional data were collected at the time of diagnosis (Table 1). 

Based on their annual rate of decline in forced vital capacity percent (FVC%) predicted, patients were categorized as slow (<10%) or rapid progressors (≥10%). The study was performed in accordance with the Declaration of Helsinki and approved by the Ethics Committee of the University Hospital of Padova (4280/AO/17). Informed consent was obtained for all study participants.

### 2.2. Study Design and Radiological Analysis

A HRCT was obtained at diagnosis (HRCT_1_) in all patients. Twenty-one patients had a second HRCT (HRCT_2_), after a median of 17 months of follow-up. The clinical and functional data of this subgroup are shown in Appendix A. HRCT_1_ and HRCT_2_ were scored blindly and independently by two expert thoracic radiologists by using a quantitative scale, as previously described [8]. Briefly, this score consists of the assessment of ground glass opacities (GGO) (alveolar score, AS%) and fibrotic extent (interstitial score, IS%) for each lung lobe. After each individual lobe was scored for both IS and AS, the final result was expressed as mean value of the five lobes for the whole lung and in different lung regions (upper and lower). The inter-observer agreement between the two radiologists was good (Cohen’s kappa 0.7), a value similar to that reported in previous studies [11].

In the twenty-one IPF patients in whom a second HRCT was available, we studied the correlation between radiological changes and FVC% decline by calculating the change of Alveolar Score (ΔAS/month), the change of Interstitial Score (ΔIS/month) and the change in FVC (ΔFVC mL/month) in the period from HRCT_1_ to HRCT_2_. We expressed the radiological changes per month to normalize the differences in timing between HRCT_1_ and HRCT_2_ in the slow and rapid progressors.

### 2.3. Pathological Analysis

Thirteen of the 49 patients underwent lung transplantation during the follow up (for clinical-functional data, see Appendix A). In all cases, the presence of UIP pattern was histologically confirmed by our expert pathologist (FC) [1]. The native lungs were fixed in formalin by airway perfusion and samples from upper and lower lobes were obtained and embedded in paraffin. Sections with a thickness of 5 μm were cut and stained for histological and immunohistochemical analysis, as previously described [3]. 

Fibroblastic foci were counted in sections stained with hematoxylin–eosin and expressed as number of fibroblastic foci/mm^2^ of area examined (Figure 1). Cellular infiltrate including total leukocytes (CD45+), neutrophils, macrophages (CD68+), and total lymphocytes calculated as sum of CD4+, CD8+ T lymphocytes as well as B lymphocytes (CD20+) was identified by immunohistochemistry as previously described [3,17,18,19] (Figure 1). Each inflammatory cell type was quantified in 20 non-overlapping high-power fields per slide and expressed as cells/mm^2^ of area examined. 

In the thirteen IPF patients in whom the histological tissue and a HRCT performed at time close to the transplantation were available, we studied the correlation between the radiological changes and the cellular inflammatory infiltrate and between the radiological changes and the fibroblastic foci count. 

### 2.4. Statistical Analysis

Statistical analyses were performed as previously described [20] (Appendix A). To compare clinical and pathological data between *rapids* and *slows*, Chi square test or Fisher’s exact test and Mann–Whitney U test were used when appropriate. To evaluate the difference between HRCT_1_ and HRCT_2_, Wilcoxon analysis was performed.

Correlation coefficients between radiological, functional and pathological findings were calculated using the nonparametric Spearman’s rank method. Adjusted *p*-values for multiple comparisons were calculated using the Holm method. The inter-observer agreement between the two radiologists was evaluated by kappa statistic measure [21]. All data were analyzed using SPSS Software version 25.0 (New York, NY, US: IBM Corp. USA) *p*-values < 0.05 were considered statistically significant.

## 3. Results

### 3.1. Clinical and Radiological Characteristics at Baseline

The clinical characteristics at baseline are shown in Table 1. Most patients were males and former smokers. Thirty patients were slow and 19 rapid progressors (median annual FVC decline: 130 mL and 689 mL respectively). None of the patients was treated with antifibrotics and 60% (equally distributed between the two groups) were treated with low dose prednisone with or without azathioprine according to previous guidelines [22]. In HRCT_1_, AS was significantly greater in rapid than in slow progressors (*p* = 0.008), while IS was similar in the two groups, either in the entire lung (Figure 2) or in different lung regions, upper and lower zones (Appendix A).

To corroborate the findings observed in previous analyses, we obtained a ROC curve on Alveolar Score data in rapid and slow progressors. We found that the area under the curve was 0.72, (95% Confidence Interval 0.57–0.87; *p* = 0.008). On the other hand, in ROC curve for Interstitial Score, we did not observe any statistically significant results (95% Confidence Interval 0.35–0.67; *p* = 0.88).

### 3.2. Pathological Analysis

In the 13 patients who were transplanted (Appendix A), we quantified the lung pathology (Figure 1A–C). The number of CD20^+^ B lymphocytes, CD4^+^ and CD8^+^ T lymphocytes (considered both individually or all together as total lymphocytes) was significantly increased in rapids than in slows (Table 2). No significant difference in the number of CD45^+^, neutrophils, macrophages and fibroblastic foci was found between rapid and slow progressors.

### 3.3. Pathological-Radiological Correlations

The total number of lymphocytes/mm^2^ was positively correlated with the HRCT AS in the whole population (Figure 3).

The number of FF/mm^2^ did not correlate with the HRCT IS in rapid progressors, slow progressors, or when considering the entire population. 

### 3.4. Functional and Radiological Characteristics at Follow Up

In the 21 patients who had a follow up HRCT_2_ (Appendix A), we found that both AS and IS increased significantly over time in both groups together (Appendix A). When the patients were divided by rate of decline, IS increased over time in both slows and rapids, while AS increased significantly only in rapids (Figure 4) (Appendix A).

### 3.5. Functional-Radiological Correlations

There was a significant correlation between the functional decline, defined as ΔFVC mL/month, and the radiological changes in IS, defined as ΔIS/month, but not with ΔAS/month. However, when stratified by the rate of decline, the correlation between ΔFVC mL/month and ΔIS/month was no longer significant in the rapid or slow decliners (Figure 5). 

When the delta IS was normalized by IS at the beginning, the correlation with the delta FVC was maintained in the patient population as a whole (*p* = 0.01, r = 0.57) but not in the slow and rapid subgroups.

## 4. Discussion

High-Resolution Computed Tomography (HRCT) plays a central role in diagnosing and staging the severity of IPF [2,5,7,8,10,12]. The HRCT relevance in IPF underlines its potential usefulness as a tool to predict future disease behavior at the time of diagnosis and to design future clinical trials [12]. 

In this study, we investigated in a group of IPF patients, followed long term (prior to available anti-fibrotic treatment), whether radiological quantification of fibrotic abnormalities and ground glass opacities in HRCT at diagnosis may predict disease behavior over time. Our results showed that, in the HRCT performed at diagnosis, patients who had experienced a rapid functional decline, rapid progressors, had a higher alveolar score (AS) than slow progressors, while the extent of fibrosis (Interstitial Score, IS) was similar in the two groups. Furthermore, in a second HRCT at follow up, changes in IS over time was correlated with functional decline. 

Idiopathic Pulmonary Fibrosis is a heterogeneous disease, the outcome of which is determined by the rapidity of the longitudinal loss of forced vital capacity, a major prognostic predictor linked to an increase in mortality risk [16,23,24,25]. The diagnosis of IPF can be done in most cases by HRCT, considered a good reflection of UIP, the pathological counterpart of IPF. Based on its diagnostic accuracy, the scoring of the different abnormalities seen in HRCT has been used in an attempt to improve the prediction of IPF outcomes [7,10]. In general, these studies show an important variability in individual disease progression and an association between mortality and increased fibrosis score over time, especially when combined with changes in FVC [7,8,10,11,12,26]. 

According to the annual FVC decline greater or lower than 10%pred, two clinical IPF phenotypes have been repeatedly reported in the literature, the rapid and the slow progressors [3,4,5,27]. These phenotypes have been shown to have a distinct gene expression profile, including the activation of important pro-inflammatory pathways that may potentially play a role in disease progression of rapid decliners [4,27]. Previous studies assessing the value of HRCT in the prediction of IPF behavior did not differentiate their patients by the predetermined rate of progression, rapid or slow, a factor of crucial significance for the understanding of disease prognosis. In the present study, we analyzed a group of patients naïve to antifibrotics classified as slow or rapid progressors, based on FVC decay over time, with the aim to identify HRCT features that could possibly differentiate at presentation rapid from slow decliners, a major prognostic predictor in IPF. 

Differently from previous studies [6,7], we evaluated not only the degree of interstitial score (IS) but also the degree of ground glass opacities, alveolar score (AS). Although pure ground glass opacity is not usually a feature of UIP, many patients with fibrotic lung disease have ground glass opacity admixed with reticular abnormality and/or traction bronchiectasis. In this context, the ground glass opacity should be regarded as part of the fibrotic process, as indicated by the recent Fleischner Society white paper [5] and, as such, we believe it needed to be assessed [2]. The HRCT findings in our study, showing that at baseline rapid progressors had an AS significantly greater (almost double) than slow progressors, is of high interest since it might help to identify, early in the course of the disease, the more aggressive phenotype with worse prognosis. 

The significance of ground glass opacities in IPF is not clear, but it might be related to parenchymal inflammatory/exudative infiltrates, probably more evident in cases with more aggressive disease and rapid progression. In support of this possibility are our recently published findings that the different clinical course (rapid or slow) of an IPF population undergoing lung transplantation was associated with distinct underlying pathology in the explanted lungs [3]. Rapid progressors showed an extensive cellular immune infiltrate, both innate and adaptive, more prominent than slow progressors [3]. The possibility that AS might represent an alveolar inflammatory/exudative infiltrate is supported by the correlation observed in the present study between AS and the total number of lymphocytes in the 13 explanted lungs. Because the timing to the second HRCT was different in the 21 patients with two consecutive HRCTs, to compare the results, we calculated the changes over time in IS, AS, and FVC and expressed them as change/per month. As suggested by others [26], the change in FVC/month in the whole group (rapids and slows together) correlated significantly with IS change/month.

Of interest and fitting with our previous findings, the AS, plausibly representing a cellular/exudative inflammatory response, increased significantly at follow up in rapid progressors while it remained stable in slow progressors. The exact explanation of this feature needs to be elucidated, however we can speculate that the AS signals a more exudative and thus unstable disease, rich in fibroblast foci, and more likely to rapidly progress towards fibrotic changes and consequently rapid functional worsening. These findings seem in line with our previous evidence on lung pathology that showed the presence of an intense lung immune infiltrate in the rapid progressors, but not in the slow progressors [3]. These results would support a role of inflammation and of adaptive immune response in determining disease behavior [28] and might account, at least in part, for the different responses to anti-fibrotic drugs among IPF patients. In favor of this possibility is our recent report showing that pirfenidone reduces FVC decline in IPF patients with a more pronounced beneficial effect in rapid than in slow progressors [29].

A strength of our study is the unique opportunity to investigate a population of IPF patients naïve of anti-fibrotic treatment, followed for at least one year, in which the disease decline phenotype, rapid or slow, could be determined by changes in FVC. Knowing the rate of decline allowed us to investigate if patterns of HRCT abnormalities could separate the rapid from slow phenotype, since the rate of FVC decay signals the worse prognosis of the disease and the response to treatment. Another important and unique feature of the study is that both AS and IS, radiological parameters which correlated with FVC decay, had a pathological confirmation. Due to the availability of effective anti-fibrotic drugs [30,31], studies on IPF patients naïve of antifibrotics will become progressively less common, if at all possible (and ethical). Thus, the HRCT patterns described could help to predict the long-term disease behavior and prognosis and be the bases for further studies in treated patients.

A limitation of our study is the relative low number of cases and the fact that the time interval between HRCT_1_ and HRCT_2_ was not the same in all patients. We corrected for this difference in timing by expressing AS and IS as changes per month. Since the pulmonary function tests were performed at the same time of HRCT, we could correlate the radiological changes with the functional changes. The quantitative estimation of HRCTs disease extent was independently evaluated by two expert radiologists with a good inter-observer agreement. It may appear a limitation not having used the automated quantitative imaging analysis [32,33,34], however visual lung scores are and have been used as “gold standard” to validate software analyses [11,35] that, in any case, have themselves some limitations and disadvantages such as the applicability to retrospective CT dataset [34,36].

## 5. Conclusion

In conclusion, quantitative estimation of High-Resolution Computed Tomography alveolar (AS) and interstitial (IS) scores reflects the distinct clinical and pathological behavior of Idiopathic Pulmonary Fibrosis slow and rapid decliners. Furthermore, the alveolar score, which reflects the immune/inflammatory infiltrate found in lung tissue, could be a useful tool to differentiate rapid from slow progressors at presentation.

## Figures and Tables

**Figure 1 jcm-08-00399-f001:**
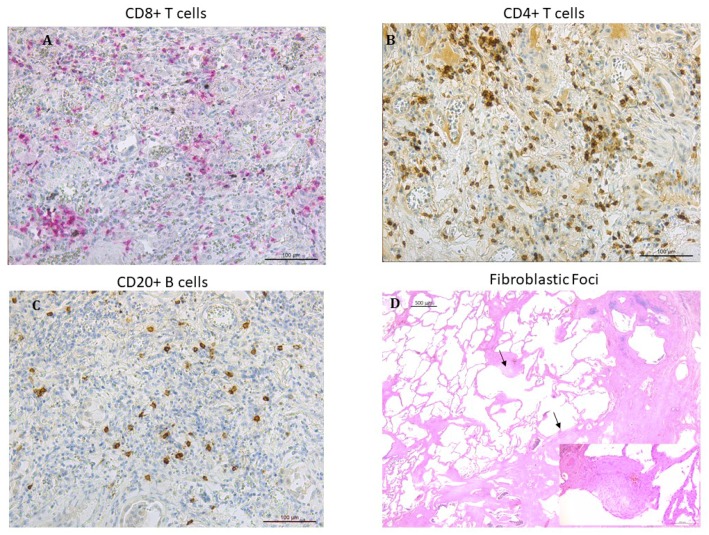
Microphotographs showing lymphocytes infiltrating the lung tissue in patients with IPF: (**A**) CD8^+^ T lymphocytes stained in red (**B**) CD4^+^ T lymphocytes stained in brown. (**C**) CD20^+^ B lymphocytes stained in brown. Immunostaining with monoclonal antibodies anti-human CD8, anti-human CD4 and anti-human CD20. (**D**) fibroblastic foci (arrows) in the transition zone between the normal lung (on the left) and the dense remodeled parenchyma with microhoneycombing (on the right) (hematoxylin and eosin staining). Insert at higher magnification: detail showing a fibroblastic focus composed of spindle cells with overlying hyperplastic pneumocytes (hematoxylin and eosin staining).

**Figure 2 jcm-08-00399-f002:**
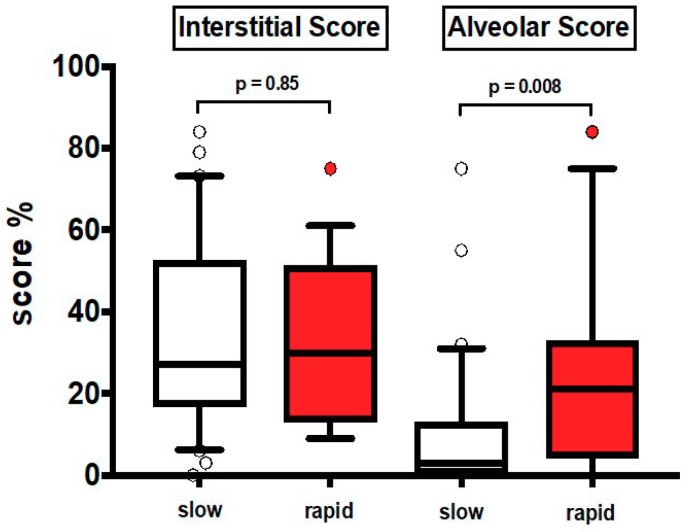
Values of HRCT_1_ Interstitial Score and Alveolar Score at baseline in slow progressors (slow) and rapid progressors (rapid). Horizontal bars represent median values; bottom and top of each box plot 25th and 75th; brackets show 10th and 90th percentiles; and circles represent outliers. White boxes indicate slow progressors and red boxes rapid progressors.

**Figure 3 jcm-08-00399-f003:**
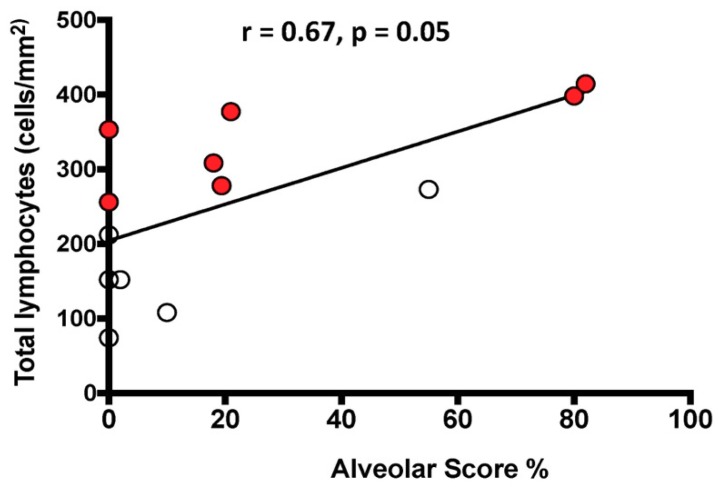
Relationship between the number of total lymphocytes infiltrating the lung tissue and the HRCT Alveolar Score. The black line represents the correlation in the entire population. White circles indicate slow progressors and red circles rapid progressors. Spearman’s rank correlation: r = 0.67, *p* = 0.01 in the entire population; r = 0.33, *p* = 0.48 in slow progressors alone; r = 0.81, *p* = 0.03 in rapid progressors alone.

**Figure 4 jcm-08-00399-f004:**
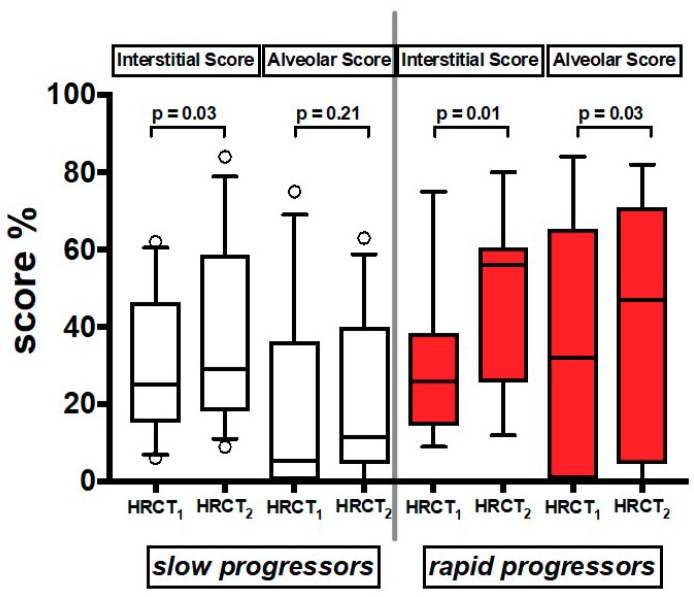
Values of Interstitial Score and Alveolar Score of the two serial HRCT scans (HRCT_1_ at baseline and HRCT_2_ at follow up). Horizontal bars represent median values; bottom and top of each box plot 25th and 75th; brackets show 10th and 90th percentiles; and circles represent outliers. White boxes indicate slow progressors and red boxes rapid progressors.

**Figure 5 jcm-08-00399-f005:**
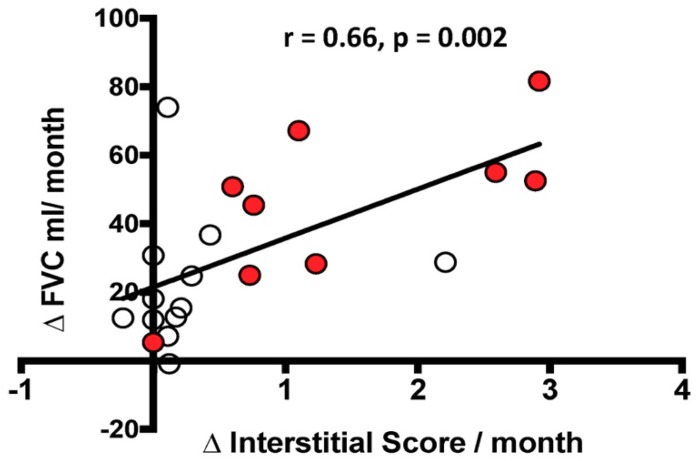
Relationship between the change over time in FVC mL (ΔFVC mL/month) and the change over time in Interstitial Score (ΔInterstitial Score/month). The black line represents the correlation in the entire population. White circles indicate slow progressors and red circles represent rapid progressors. Spearman’s rank correlation: r = 0.66, *p* = 0.002 in the entire population; r = 0.31, *p* = 0.6 in slow progressors alone; r = 0.73, *p* = 0.06 in rapid progressors alone.

**Table 1 jcm-08-00399-t001:** Demographics and clinical characteristics of the entire population (n = 49), including 30 slow and 19 rapid progressors.

	Entire Population (n = 49)	*Slow Progressors* (n = 30)	*Rapid Progressors* (n = 19)	*p* Value
Male *–* n (%)	42 (86)	24 (80)	18 (94)	0.22
Age at diagnosis *–* years	58 (33–74)	58 (46–74)	60 (33–69)	0.75
Smoking history *–* pack years	20 (0–93)	15 (0–60)	21 (0–93)	0.24
• Current *–* n (%)	2 (4)	1 (3)	1 (5)	1
• Former *–* n (%)	40 (82)	23 (77)	17 (89)	0.45
• Non smokers *–* n (%)	7 (14)	6 (20)	1 (5)	0.22
Symptoms duration at diagnosis – months	20 (0–240)	20 (0–240)	18 (0–120)	0.58
Radiological diagnosis *–* n (%)	28 (57)	20 (67)	8 (42)	0.13
FVC at diagnosis *–* L	2.34 (1.19–4.06)	2.18 (1.19–4.06)	2.51 (1.75–4)	0.38
FVC at diagnosis *–* %pred.	67 (36–109)	66 (36–109)	76 (46–107)	0.52
DL_CO_ at diagnosis *–* %pred.	47 (10–97)	45 (25–97)	50 (10–82)	0.73
FVC decline per year *–* mL	275 (−330–1498)	130 (−330–380)	689 (331–1498)	<0.0001
FVC decline per year *–* %pred.	9 (−30–35)	4 (−30–9)	16 (11–35)	<0.0001
Patients undergoing transplant *–* n (%)	13 (27)	6 (20)	7 (37)	0.31
Patients who died *–* n (%)	28 (57)	15 (50)	13 (68)	0.2

Values are expressed as numbers and percent or median and ranges. Negative values mean improvement of FVC. *p*-values refers to comparison between slow and rapid progressors.

**Table 2 jcm-08-00399-t002:** Inflammatory cells numbers of the entire population with lung pathology (n = 13), including six slow and seven rapid progressors.

	Entire Population (n = 13)	*Slow Progressors* (n = 6)	*Rapid Progressors* (n = 7)	*p* Value
Total leukocytes CD45^+^ -, cells/mm^2^	352 (149–732)	284 (149–383)	379 (333–732)	0.7
Macrophages, cells/mm^2^	136 (63–308)	132 (63–308)	136 (71–303)	0.9
Neutrophils, cells/mm^2^	51 (2–138)	6 (2–62)	51 (4–138)	0.1
Total lymphocytes, *cells/mm*^2^ • CD 20^+^ B lymphocytes • CD 4^+^ T lymphocytes • CD 8^+^ T lymphocytes	273 (74–414) 42 (25–115) 138 (20–284) 44 (12–120)	152 (74–273) 36 (27–115) 87 (20–138) 33 (12–45)	353 (256–414) 62 (25–115) 194 (115–284) 66 (26–120)	0.002 0.008 0.002 0.001
Fibroblastic foci, n/mm^2^	2.7 (1–7)	2.8 (2–7)	2 (1–4.6)	0.09

Values are expressed as median and ranges. *p* values refers to comparison between slow and rapid progressors.

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
