# Peer review of "High-Resolution Computed Tomography (HRCT) Reflects Disease Progression in Patients with Idiopathic Pulmonary Fibrosis (IPF): Relationship with Lung Pathology"

_jcm, 2019, doi:10.3390/jcm8030399_

Reviewer 1 Report

In this manuscript Cocconcelli et al. assess the potential usefulness of ground glass and fibrotic extension to predict the clinical course of IPF. They prospectively compared HRCT imaging to lung function as well as fibrotic extension in lungs. The manuscript is well written and the language is good. The clinical question is of interest for those who deal with IPF and the scientific exploration original. Nevertheless I’ve some concern about this manuscript.

 Major :

 -       How does the authors explain the increase of FVC of 30% ! in some slow progressor.

-       How many patients underwent a lung biopsy? Focusing on old guidelines (2011) patients with GGO have to be confirmed IPF with SLB. This is a major outcome, which has to be solved to confirm the conclusions of the authors. Of course, patient who had benefit from lung Tx have been confirmed, what about the others?

-       What is the rationalized method to analyze histopathology and to quantify the lesions?

-       What about the rate of acute exacerbations ant the time to AE from the first HRCT (prognostic?)

-       How many anatomopathologist did confirm the histopathologic score? The correlation between IS and FF is very high. How do the authors explain that new observation despite previous studies focusing on that specific topic? Could the authors discuss that observation in the discussion and underlies the potential limitation due to the low number of patients?

 Minor :

- Do the authors correlate GGO/HRCT score with balf lymphocytosis or balf with anatomopathology

- What are the median time between 2 HRCT and the median time between the histopathological analysis and the HRCT (with detail in annex for each patients)

- please provide the ethic committee ref.

- the time to Tx is very short according to the mean European waiting time, how can the authors explain that observation? (previous diagnosis in  other center?)

Author Response

Response to Reviewer 1 Comments

 Major:

Point 1: How does the authors explain the increase of FVC of 30%! in some slow progressor.

Response 1. The referee is correct. However, the significant FVC increase he/she is alluding to is due to a single patient (i.e., an outlier) who actually experienced such level of functional improvement. This patient clinical notes have been carefully reviewed and the functional data double checked. We thank the referee for bringing this up.

Point 2: How many patients underwent a lung biopsy? Focusing on old guidelines (2011) patients with GGO have to be confirmed IPF with SLB. This is a major outcome, which has to be solved to confirm the conclusions of the authors. Of course, patient who had benefit from lung Tx have been confirmed, what about the others?

Response 2: We thank the reviewer for his/her important comment. Indeed, according to the 2011 guideline document, patients with a radiological (i.e., HRCT) pattern of possible UIP had to undergo histological confirmation of their diagnosis. This was actually the case for 18/36 individuals who did not undergo lung transplant. In the remaining cases (18/36), the diagnosis was made based on clinical and HRCT features, functional decline over time as well as exclusion of all known causes of pulmonary fibrosis (i.e., negative autoimmunity, and BAL and transbronchial biopsy not suggestive of alternative diagnoses). Notably, in this latter patient subset, 7 patients refused to undergo surgical lung biopsy while in 9 the procedure was contraindicated due to major comorbidities.

Point 3: What is the rationalized method to analyze histopathology and to quantify the lesions?

Response 3: The referee highlights an important point. Details about the methodology used to analyze histopathology specimens and to quantify the lesions are missing in the manuscript and we apologize for this. The methods section has now been amended as appropriate (please see page 4, Pathological analysis).

 Point 4: What about the rate of acute exacerbations and the time to AE from the first HRCT (prognostic?)

Response 4: The rate of acute exacerbations was 12% (6/49) and the median time (months) between HRCT and the occurrence of the exacerbation was 15 months. Overall, the HRCT features (ground glass opacity, reticulation and honeycombing) were not predictive of the risk of exacerbation, although the number of patients is too small to draw firm conclusions.

Point 5: How many anatomopathologist did confirm the histopathologic score? The correlation between IS and FF is very high. How do the authors explain that new observation despite previous studies focusing on that specific topic? Could the authors discuss that observation in the discussion and underlies the potential limitation due to the low number of patients?

Response 5: Two experienced thoracic pathologists (FC and FP) performed the histopathologic examination blindly and independently and controversies were solved by consensus. With regard to the referee’s comment about the strong correlation between IS and FF, actually after adjustment by using the Holm method, as requested by the second referee, the association between FF profusion and IS (i.e., reticular changes) was no longer significant neither in the patient population as a whole nor in rapid progressors. Accordingly, we decided to remove Figure 4 (the numbering refers to the original submission), and we hope the referee is happy with this.

Minor:

Point 6: Do the authors correlate GGO/HRCT score with balf lymphocytosis or balf with anatomopathology.

Response 6: BAL cell count was available for only a small proportion of our patients. Therefore, such analysis would be unlikely to provide meaningful results.

Point 7: What are the median time between 2 HRCT and the median time between the histopathological analysis and the HRCT (with detail in annex for each patients).

Response 7:  The time between HRCT1 and HRCT2 was 17 (5-87) months - expressed as median and range - whereas the time between the most recent HRCT and transplantation was 1 (0-20) month - expressed as median and range. These data are also available in the Supplementary material (Table S1 and Table S2, respectively).

Point 8: Please provide the ethic committee ref.

Response 8:  The Ethics Committee reference number is 4280/AO/17 - University Hospital of Padua.

Point 9: The time to Tx is very short according to the mean European waiting time, how can the authors explain that observation? (previous diagnosis in other center?)

Response 9: We apologize for the confusion and thank the referee for highlighting this point. We believe the “short time” period the reviewer is alluding to refers to the time between the most recent HRCT available and transplantation, which is a slightly different concept. In fact, in our study, the waiting time in list for lung transplant was in line with that of other transplant centers across Italy (i.e., approximately 18 months based to blood group). As correctly hypothesized by the referee, most transplanted patients in our study were originally diagnosed with IPF in other centers.

Reviewer 2 Report

This study explores the potential role of HRCT to predict outcome in IPF and relates CT phenotype to disease progression rate in a cohort of patients from two centres. CT imaging was scored using a visually-based scale to estimate the extent of two key characteristics, ground glass opacities and fibrosis, and related to disease progression defined using decline in FVC. In a sub-group of patients who underwent lung transplantation CT appearance was correlated with histopathological appearance. The authors report that baseline alveolar score was greater in patients who subsequently showed more rapid decline in FVC. In those patients who underwent lung transplantation, some histopathological characteristics related to decline rate and to CT appearance.

In a further subgroup of 21 subjects who underwent interval CT imaging, the fibrosis score worsened over time and, in rapid decliners, the ground glass change progressed significantly. In rapid decliners, there was a correlation between decline in FVC and worsening fibrosis score.

 The authors conclude that

 Major comments:

The study was not observational but patients received therapy that might have modified the phenotype or disease progression, particularly the presence of ground glass change. I could not see any reference to this in the results or subgroup analysis comparing patients with and without treatment and no reference to this potentially confounding factor in the discussion. This is particularly pertinent since the numbers are quite small (as expected in a study of IPF) and the strength of correlations and relationships quite weak. Consequently, there may be a significant influence of treatment on the results.

Please explain why none of the patients were commenced on anti-fibrotic medication when it was available.

There are a number of semi-automated tools to assess ILD but the authors did not incorporate them in their study. Please could you justify why you did not include the latest methodology?

The authors suggest that alveolar score could be a useful tool to differentiate rapid from slow progressors at presentation, presumably in routine clinical practice. In the absence of data using an automated programme, I think that it may be impractical to suggest the use of visual quantitative assessment in routine clinical practice and this statement should be qualified. I would also wish to see data indicating the positive predictive value of AS in predicting rapid progression since this would demonstrate whether it is justifiable to state that the method represents a useful tool for the proposed purpose.

 Many of the correlations are with very small numbers and the relationships are not particularly convincing. Was any adjustment made for multiple statistical testing?

 The interval between HRCT 1 and HRCT2 is very varied and differed between slow and rapid progressors. It is not clear what influence the varied interval between CTs might have had on the data and the relationships demonstrated, and how the difference in interval between rapid and slow progressors arose. Please would you discuss these issues, giving an explanation of how these situations arose and the potential influence on the results.

 The scores were performed for individual lobes and averaged. Please would the authors include analyses for the different regions of the lung to demonstrate whether distribution of disease is an important phenotypic characteristic and predictor of progression.

 Minor comments:

Please would the authors explain the statement ‘It is important to recognize that a HRCT IS% of less than 25% in all lobes identifies patients with only reticulations on HRCT, whereas an HRCT IS% of more than 25% identifies patients with both reticulations and honeycomb changes in different degree.’

 Author Response

Response to Reviewer 2 Comments

 Major comments:

Point 1: The study was not observational but patients received therapy that might have modified the phenotype or disease progression, particularly the presence of ground glass change. I could not see any reference to this in the results or subgroup analysis comparing patients with and without treatment and no reference to this potentially confounding factor in the discussion. This is particularly pertinent since the numbers are quite small (as expected in a study of IPF) and the strength of correlations and relationships quite weak. Consequently, there may be a significant influence of treatment on the results.

Response 1: We thank the reviewer for his/her comment and apologize for not making this point sufficiently clear. Sixty percent of patients (equally distributed between the two groups) were treated with low dose prednisone with or without azathioprine. Moreover, patients were all free of antifibrotic treatment. Therefore, it is unlikely that differences in disease progression between the two groups are due to a between group imbalance in treatment strategies.

Point 2: Please explain why none of the patients were commenced on anti-fibrotic medication when it was available.

Response 2:  Patients were all free of antifibrotic treatment simply because - for the purpose of our study - we only considered functional and radiological data up until the start of antifibrotic treatment. However, a minority of our patients belonged to an historical cohort from the pre-antifibrotic therapy era (before 2014). This is now clarified in the result section, page 6, first paragraph.

Point 3: There are a number of semi-automated tools to assess ILD but the authors did not incorporate them in their study. Please could you justify why you did not include the latest methodology?

Response 3: Semi-automated tools to assess disease extent and severity in patients with fibrotic ILD is not (yet) available in our Institution. Similarly, in many ILD centers, including expert referral centers, such tools are still not available, at least in routine clinical practice, whereas they are largely used in research settings. However, we specifically wanted to conduct a study as close as possible to the reality of daily practice in which a visual analysis of CT images is generally performed.  Wu and colleagues, in a recent elegant review article, (reference 36) described advantages and disadvantages of quantitative computed tomography (QCT) and highlighted that a computer-based evaluation has a number of inherent boundaries that may restrict the utility of QCT, particularly in retrospective CT datasets.

Point 4: The authors suggest that alveolar score could be a useful tool to differentiate rapid from slow progressors at presentation, presumably in routine clinical practice. In the absence of data using an automated programme, I think that it may be impractical to suggest the use of visual quantitative assessment in routine clinical practice and this statement should be qualified. I would also wish to see data indicating the positive predictive value of AS in predicting rapid progression since this would demonstrate whether it is justifiable to state that the method represents a useful tool for the proposed purpose.

Response 4:  We thank the reviewer for these comments. Regarding visual assessment, please see response to comment 3. Regarding the positive predictive value of AS the referee is correct. To improve clarity of the results, as suggested, we obtained a ROC curve analysis on Alveolar Score data in rapid and slow progressors. We found that the area under the curve, in the ROC analysis was 0.72, (95% Confidence Interval 0.57 - 0.87; p=0.008) strongly suggesting that Alveolar Score may represent a promising predictor of disease progression, although this finding needs to be validated in larger populations of prospectively enrolled patients. On the other hand, in ROC curve analysis for Interstitial Score we did not observe any statistically significant results (95% Confidence Interval 0.35 - 0.67; p=0.88). This is clarified in the Results, page 6.

Point 5: Many of the correlations are with very small numbers and the relationships are not particularly convincing. Was any adjustment made for multiple statistical testing?

Response 5: We thank the reviewer for raising this important point. The issue of whether adjustments should be applied in studies such as the present one when multiple comparisons are performed is highly debated. Some authors favor the more stringent Bonferroni method, while others argue that this approach is too conservative and may actually mask real associations. Therefore we decided to apply the less conservative Holm correction (Aickin M, Gensler H. Adjusting for multiple testing when reporting research results: the Bonferroni vs Holm methods. Am J Public Health 1996; 86: 726-8. Curtin F, Schulz P. Multiple correlations and Bonferroni’s correction. Biol Psychiatry. 1998; 44: 775-7). After a Holm correction for multiple testing, in the entire patient population, the correlations still held, in particular the association between the number of tissue lymphocytes and Alveolar Score (Page 7, Figure 3) and the association between the change in FVC and the change in Interstitial Score over time (Page 9, Figure 6 of the original manuscript).  Conversely, after Holm correction, the correlation between FF/mm2 and HRCT IS was no longer significant. Therefore, we have decided to remove Figure 4 in the original submission.

Point 6: The interval between HRCT1 and HRCT2 is very varied and differed between slow and rapid progressors. It is not clear what influence the varied interval between CTs might have had on the data and the relationships demonstrated, and how the difference in interval between rapid and slow progressors arose. Please would you discuss these issues, giving an explanation of how these situations arose and the potential influence on the results.

Response 6:  The reviewer is correct and we apologize for not clarifying this point sufficiently. The differences in time interval between HRCT1 and HRCT2 are mainly driven by clinical conditions; in other words, in the presence of rapid clinical-functional deterioration, rapid progressors were more likely to undergo a follow-up CT than slow progressors. Indeed, to normalise the differences in timing between HRCT1 and HRCT2 in the slow and rapid progressors, we expressed the radiological changes per month.

This has now been clarified. Please see also the Methods section, Study design and Radiological analysis, page 4.

Point 7: The scores were performed for individual lobes and averaged. Please would the authors include analyses for the different regions of the lung to demonstrate whether distribution of disease is an important phenotypic characteristic and predictor of progression.

 Response 7: We thank the referee for this comment. We have now performed the analysis he/she suggested, and summarized the results in the manuscript and in Table S3 of the supplementary material. In HRCT1, Alveolar Score (AS), assessed either in different lung regions (upper and lower) or in the entire lung, was significantly greater in rapid than in slow progressors as explained in the revised manuscript (page 7, second paragraph and in Table S3). With regard to Interstitial Score (IS), in HRCT1, measured either in different lung regions (upper and lower) or in the entire lung, it was similar between rapid and slow progressors as expressed in the revised manuscript (page 6, second paragraph and Table S3 of the supplementary material).

Minor comments:

Point 8: Please would the authors explain the statement ‘It is important to recognize that a HRCT IS% of less than 25% in all lobes identifies patients with only reticulations on HRCT, whereas an HRCT IS% of more than 25% identifies patients with both reticulations and honeycomb changes in different degree.’

Response 8: The sentence the reviewer is alluding to refers to the quantification of Interstitial Score (IS) as originally described by Fell and colleagues (Fell CD et al. Am J Respir Crit Care Med 2010; 181: 832-7). However, we agree that the sentence may lend itself to some misunderstanding and we decided to remove it from the manuscript.

 Round  2

Reviewer 1 Report

Functional & radiological correlations : we cannot speak about trend in statistical analysis, this is not correlated, therefore we cannot consider anything else. Please modify the sentence.

Could the author consider to normalize the evolutivity of the IS and AS (delta) / IS or AS score at the beginning. Could we hypothetize to correlate withe PFT.

Author Response

Response to Reviewer  1

 Point 1: Functional & radiological correlations: we cannot speak about trend in statistical analysis, this is not correlated, therefore we cannot consider anything else. Please modify the sentence.

 Response 1: The reviewer is correct. Therefore, the sentence he/she is alluding to has now been removed from the Results section (page 9). A similar sentence referring to the same point has been also removed from the Discussion (page 11).

 Point 2: Could the author consider to normalize the evolutivity of the IS and AS (delta) / IS or AS score at the beginning. Could we hypothetize to correlate withe PFT.

 Response 2. The additional analysis the referee is alluding to has now been performed. While the correlation between delta AS / AS at the beginning and delta FVC was not significant neither in the entire patient population (p=0.46, r=0.19) nor in the slow (p=0.91, r= - 0.05), and rapid (p=0.66, r=0.21) subsets, the correlation between delta IS / IS at the beginning and delta FVC was maintained in the patient population as a whole (p=0.01, r=0.57) but not in the slow and rapid subgroups (p=0.9, r=0.04 and p=0.06, r=0.65, respectively).

These findings have now been included in the Results section (page 9).

Reviewer 2 Report

Thank you for your responses.

The manuscript is significantly improved. However, it remains unclear whether anti-fibrotic medication  would have been available to patients at the time of the study. The response to Point 2 suggests that the majority of patients were not in a cohort that was pre-antifibrotic therapy and would therefore have missed the opportunity to receive disease modifying treatment. From an ethical standpoint there needs to be absolute clarity on this point.

 Response to Point 6. By choosing an interval between scans that was based on decline rate, there is a potential to introduce bias into the monthly  radiology changes and this also assumes a linear rate of decline.

Author Response

Response to Reviewer  2

 Point 1: The manuscript is significantly improved. However, it remains unclear whether anti-fibrotic medication  would have been available to patients at the time of the study. The response to Point 2 suggests that the majority of patients were not in a cohort that was pre-antifibrotic therapy and would therefore have missed the opportunity to receive disease modifying treatment. From an ethical standpoint there needs to be absolute clarity on this point.

 Response 1: We apologize for not making this important point clear enough. All patients in our study, whether from our center or referred to our center, were offered antifibrotic therapy as soon as it became available, provided they met the FVC, DLCO and age criteria for treatment and they had no clear contraindications to it. However, the aim of our study was to look at a population of patients off treatment, which is why we considered only radiological and functional data before antifibrotic therapy was instituted.

This said, a minority of our patients belonged to an historical cohort from the pre-antifibrotic therapy era (before 2014) and they had no access to antifibrotic therapy.

This is now clarified in the method section, study population, page 3.

 Point 2: Response to Point 6. By choosing an interval between scans that was based on decline rate, there is a potential to introduce bias into the monthly radiology changes and this also assumes a linear rate of decline.

 Response 2: We are not sure we understand this comment correctly. The different time interval between CTs in rapid and slow progressors was not prespecified but driven by clinical conditions.
What we wanted to look at was if lung function change was related in any way to change in CTs, which were done only for patient care; this accounts for the different intervals between scans. In order to normalize for different intervals among subjects, we decided to express CT changes as delta IS/month and delta AS/month. Because the intervals between scans were at least one year (and often longer), we believe the delta IS/month and delta AS/month could realistically reflect the real decline over time.